# Mitochondria Specific Antioxidant, MitoTEMPO, Modulates Cd Uptake and Oxidative Response of Soybean Seedlings

**DOI:** 10.3390/antiox11112099

**Published:** 2022-10-25

**Authors:** Dalir Fayazipour, Joanna Deckert, Gholamali Akbari, Elias Soltani, Jagna Chmielowska-Bąk

**Affiliations:** 1Department of Agronomy and Plant Breeding Sciences, College of Aboureihan, University of Tehran, Tehran P.O. Box 3391653775, Iran; 2Department of Plant Ecophysiology, Institute of Experimental Biology, Faculty of Biology, School of Natural Sciences, Adam Mickiewicz University, 61-712 Poznań, Poland

**Keywords:** metal stress, oxidative stress, 8-hydroxyguanosine, 8-hydroxyguanine

## Abstract

Numerous reports find that Cd induces formation of reactive oxygen species (ROS) in plants. However, a general ROS pool is usually studied, without distinction of their production site. In the present study, we applied a mitochondria-specific antioxidant, MitoTEMPO, to elucidate the role of mitochondria-derived ROS in the response of soybean seedlings to short-term (48 h) Cd stress. The obtained results showed that Cd caused a reduction in root length and fresh weight and increase in the level of superoxide anion, hydrogen peroxide, markers of lipid peroxidation (thiobarbituric reactive substances, TBARS) and markers of RNA oxidation (8-hydroxyguanosine, 8-OHG) in seedling roots. Application of MitoTEMPO affected Cd uptake in a dose-dependent manner and diminished the Cd-dependent induction of superoxide anion and lipid peroxidation.

## 1. Introduction

Contamination of the environment with metals, including cadmium (Cd), is a serious concern worldwide. Cadmium is a non-essential metal, toxic even in relatively low concentrations. It is classified in the seventh position on the Substances Priority List published by the Agency for Toxic Substances and Disease Registry, which comprises chemical compounds posing the greatest threat to human health [1] Cadmium is also toxic to plants and the effects of its toxicity include growth inhibition, decreased photosynthesis efficiency, altered mineral homeostasis and genotoxicity [2]. These symptoms are, at least in part, associated with oxidative stress. Over-production of reactive oxygen species (ROS) and an increase in the level of oxidative stress markers, such as thiobarbituric acid reactive substances (TBARS) and malondialdehyde (MDA), are common effects of plant exposure to Cd [2,3]. Due to their high reactivity, ROS adversely affect plant cells through, among other processes, peroxidation of lipids, carbonylation of proteins and oxidation of nucleic acids, leading to altered structure and functioning of these biomolecules. The exact effects depend on the type of reactive oxygen species, its concentration and localizations. In general, hydroxyl radical (^·^OH) is perceived as the most reactive and deleterious ROS, followed by superoxide anion (O_2_^·−^) and singlet oxygen (^1^O_2_). Among various ROS types, hydrogen peroxide (H_2_O_2_) is recognized as the most long lasting but simultaneously shows relatively low reactivity [4,5]. Despite their toxicity at high concentration, it is also generally acknowledged that at certain levels ROS are indispensable for proper cell functioning through participation in, among other processes, cellular and long distance signaling, regulation of developmental processes and stress response [6]. 

There are various sites of ROS generation in plants including chloroplasts, mitochondria, peroxisomes, membranes and apoplast. In chloroplasts and mitochondria, ROS are generated in electron transport chains. In chloroplasts, altered equilibrium between light harvesting and energy production may additionally lead to the formation of singlet oxygen. In peroxisomes, ROS can be formed, e.g., as a result of β-oxidation of fatty acids and conversion of xanthine and hypoxanthine to uric acid via the action of xanthine oxidase. In the case of apoplast, the main ROS-producing sites include apoplastic peroxidases and membrane bond enzyme—NAPDH oxidase (also referred to as respiratory burst oxidase homolog proteins, RBOH) [5,7]. So far, little is known about the exact sources of ROS in plant response to Cd. The studies on tobacco suspension cells revealed three waves of early ROS generation. The first wave depended on the activity of NADPH oxidase, the second one consisted of superoxide anion (O_2_^·−^) generation in mitochondria, while the third wave included accumulation of fatty acid peroxides [8]. Similarly, two waves of ROS were observed in soybean response to Cd. The first wave depended mostly on hydrogen peroxide (H_2_O_2_) and nitric oxide (NO) generation, while the second one consisted of O_2_^·−^ generation. The first wave was accompanied by a slight increase in NADPH oxidase activity pointing out the possible engagement of this enzyme in ROS generation [9]. The putative involvement of NADPH oxidase in soybean response to Cd is further supported by studies applying the pharmacological approach. The application of DPI, a commonly used NADPH oxidase inhibitor, resulted in diminished production of O_2_^·−^ in reaction to short-term Cd treatment [10]. In turn, studies on Arabidopsis protoplasts pointed to mitochondria as one of the first ROS-producing sites in response to Cd. The results of the study revealed significant ROS accumulation in mitochondria already within the first hour of Cd application [11].

The research on the effects of ROS generated in specific cellular compartments is facilitated by the introduction of new, organelle targeting antioxidants, such as MitoTEMPO. This compound consists of antioxidant, piperidine nitroxide, bound to triphenylphosphonium (TPP), which facilitates its passing through biological membranes and action inside of the mitochondria [12]. MitoTEMPO has been applied in numerous studies in animal models, showing among other uses, its possible utilization in the treatment of cancer, neurodegenerative diseases and cardo-vascular disorders [12,13,14,15]. For instance, application of MitoTEMPO to cardiomyocytes isolated from aged Wistar rats resulted in a decrease in the level of superoxide anion in mitochondria, improved mitochondria ultrastructure and diminished levels of apoptosis markers. Additionally, the compound exhibited anti-arrhythmic effects through regulation of Ca^2+^ levels [16]. In adipose-derived stem cells (ADSCs) isolated from diabetic mice treatment with MitoTEMPO improved cell proliferation, multidifferentiation and proangiogenic potential. Importantly, the same effects were not observed by application of solely TEMPO, which is not targeting the mitochondria [17]. One of the symptoms of Alzheimer’s Disease is deposition of amyloid beta (Aβ). Research on primary cultured mice neurons revealed that application of MitoTEMPO reversed some of the adverse effects of Aβ treatment including superoxide over-production, increased lipid peroxidation and decreased cytochrome c activity and ATP production [18].

In turn, there are very limited reports on MitoTEMPO effects in plants. So far, only one study showed that its application in artificially aged elm seeds results in diminished ROS production and hampered mitochondria aggregation accompanied by an increase in the germination rate in relation to the aged seeds. Additionally, treatment with these mitochondria targeting antioxidant caused proteomic changes, which included a decrease in the abundance of proteins associated with the tricarboxylic acid (TCA) cycle, the heat shock protein 90 (HSP 90) and a protease homolog. On the other hand, an increase in the level of elongation factor 1, aldehyde dehydrogenase, ATP-dependent Clp protease and outer membrane protein VDAC has been noted [19].

The aim of the current study was elucidation of the role of mitochondria-derived ROS in the response of soybean seedlings to short-term Cd exposure. Soybean seedlings were treated with Cd with or without the addition of MitoTEMPO to examine the possible effect of the antioxidant on the development of common Cd toxicity symptoms, including growth alterations and an increase in the level of oxidative stress markers—superoxide anion, hydrogen peroxide and the products of lipid, protein and ribonucleic acids oxidation.

## 2. Materials and Methods

### 2.1. Cultivation and Treatment Procedures

Soybean (*Glycine max* L cv. Naviko) seeds were obtained from the Department of Genetics and Plant Breeding of the University of Life Sciences in Poznań, Poland. Seeds were surface sterilized with 75% ethanol for 5 min and 1% sodium hyperchlorite for 10 min. Then, the seeds were washed under running water for 30 min and soaked in tap water for 3–4 h. Around 200 seeds showing no symptoms of damage (black or brown coloring, cracking) were placed on trays with two layers of moistened lignin covered by a layer of blotting paper. The trays were covered with aluminium foil to preserve the humidity and transferred to a growth chamber. The seeds were germinated for two days (48 h) in the dark at the stable temperature of 22 °C. Thereafter, germinated seedlings, selected on the basis of similar root length, were transferred to glass Petri dishes (10 cm diameter) with the roots placed between two layers of blotting paper in cut out holes. Seedlings were then treated with 5 mL of the following solutions: distilled water (control), distilled water with 20 nM MitoTEMPO (Sigma-Aldrich, St Louis, MO, USA, SML0737), CdCl_2_ with Cd at concentrations of 10 mg/L (corresponding to 89 µM), CdCl_2_ with Cd at concentrations of 10 mg/L with 20 nM MitoTEMPO, CdCl_2_ with Cd at concentrations of 25 mg/L (corresponding to 223 µM) and CdCl_2_ with Cd at concentrations of 25 mg/L with 20 nM MitoTEMPO. Then, the Petri dishes were covered with aluminium foil and transferred to a growing chamber for an additional two days (48 h) of germination. The growth parameters, superoxide level and lipid peroxidation intensity were assessed after 48 h of treatment on fresh seedlings. For Cd quantification, the seedlings were thoroughly washed with distilled water, and the roots were cut off and dried at 60 °C. For other analyses (assessment of hydrogen peroxide level, protein carbonylation and RNA isolation), the roots of the seedlings were cut off on ice, immediately snapped frozen in liquid nitrogen and stored at −80 °C.

The applied Cd concentrations were based on previous studies carried out on soybean seedlings in similar conditions, showing that 25 mg/L Cd inhibits the root length by 50% [20]. The MitoTEMPO concentration has been chosen on the basis of published data on elm seeds [19].

### 2.2. Measurement of Growth Parameters

Root length and fresh weight of roots were measured after 48 h of treatment. The weight of the fresh roots was measured with a digital balance with an accuracy of 0.01 g. The analysis was performed in five biological replicates, each consisting of 30 roots.

### 2.3. Evaluation of Hydrogen Peroxide Level

Hydrogen peroxide content was measured using the “Hydrogen Peroxide Assay Kit” (Abcam, Cambridge, England, Ab 102500) based on an OxiRed probe with small modifications. Previously cut roots of soybean seedlings (300–400 mg), frozen in liquid nitrogen and stored at −80 °C, were homogenized in chilled mortars in liquid nitrogen. The homogenized samples were transferred to Eppendorf tubes containing 500 µL of assay buffer and centrifuged (13,000 rcf, 4 °C, for 15 min). Subsequently, 50 µL of the supernatant or provided standard were added to the microplate wells and supplemented with 50 µL of the reaction mixture containing the assay buffer, OxiRed probe and horseradish peroxidase. Due to the rapid color change, absorbance was measured immediately at λ = 570 nm on a Synergy LX microplate reader (BioTek, Wnooski, VT, USA). Each sample was measured in two technical replicates. The analysis was performed in four biological replicates.

### 2.4. Evaluation of Superoxide Anion Level

The nitrotetrazolium (NBT) test was used to determine the level of superoxide anion [21] The roots of soybean seedlings were incubated for 1 h in the dark in a reaction mixture containing 0.05 M potassium buffer, 0.05% NBT (Serva, Heidelberg, Germany, 30550.03), 0.1 mM EDTA and 10 mM NaN_3_ (Sigma-Aldrich, 71290). Thereafter, the seedlings were removed from the solution and their roots were cut off and weighed. Subsequently, the samples (solution) were incubated for 15 min at 85 °C. The absorbance of the solution was measured at λ = 580 nm. The analysis was performed in three biological replicates.

### 2.5. Measurements of Lipid Peroxidation

Lipid peroxidation was assessed by the amount of TBARS (2-thiobarbituric acid reactive substances) according to [22], with minor modifications. The roots of the soybean seedlings (300–400 mg) were cut on ice and homogenized with 2 mL of 10% TCA (trichloroacetic acid solution, Sigma- TO699). Samples were centrifuged (12,000 rpm, 4 °C, 10 min), and 1 mL of the supernatant was transferred to glass tubes. Then, the tubes were filled with 4 mL of 0.5% TBA (2-thiobarbutric acid, Sigma-T5500-25G) dissolved in 10% TCA and incubated at 95 °C for 30 min. Subsequently, the samples were cooled for 15 min on ice. The absorbance of the supernatant was measured at λ = 532 nm and corrected for the nonspecific absorbance at λ = 600 nm. The analysis was performed in six biological replicates.

### 2.6. Assessment of Protein Carbonylation

The carbonylation of the proteins was measured using the Protein Carbonyl Content Assay Kit (Merck, Rahway, NJ, USA, MAK094). The roots of soybean seedlings (300–400 mg) were homogenized in chilled mortars in 500 µL of water free from RNAses, DNAses, and proteinases (BioShop, Burlington, Canada, WAT333.500) and processed according to the provided kit manual. The absorbance was measured by λ = 375 nm on a Synergy LX microplate reader (BioTek, Wnooski, VT, USA). Pure guanidine solution was used as the negative control. Each sample was assayed in two technical replications. The protein level was measured with the use of the QuantiPro BCA Assay Kit (Sigma-Aldrich, St Louis, MO, USA, QPBCA) according to the manufacturer’s manual. Briefly, 50–100 µL of supernatant was transferred to microplate wells and supplemented with DNAase/RNase/proteinase-free water (BioShop, WAT333.500) to the total volume of 150 µL. Thereafter, each well was supplemented with 150 µL of provided working reagent. Due to rapid color change, the absorbance of the samples was measured after 30 min of incubation at 37 °C by the λ = 562 nm. Each sample was measured in two technical repetitions. The level of carbonyl groups was calculated according to the formula provided in the Protein Carbonyl Content Assay Kit manual: CP = (A375/6.364) × 100/mg of protein × 1000, wherein CP—nmol carbonyl/mg protein. The analyses were performed in 4–6 biological repetitions, each consisting of 10 roots.

### 2.7. Quantification of 8-Hydroxyguansoine (8-OHG) Level in Total RNA

For quantification of 8-hydroxyguanosine (8-OHG), which is the most frequent marker of RNA oxidation, the total RNA was isolated using TriReagent (Sigma-Aldrich, St Louis, MO, USA T9424) in sterile conditions. Briefly, 1 mL of TriReagent and 2 stainless-steel beads (Qiagen, Hilden, Germany, Cat No./ID: 69989,) were added to each sample. The samples were homogenized on TissueLyser II (Qiagen) for 3 min at 25 rounds/min. The samples were incubated for 20 min at RT, supplemented with 200 µL of chloroform (Sigma-Aldrich, 32211), thoroughly mixed, incubated for 15 min at RT and centrifuged for 20 min at 4 °C by 12 000 rpm. The aqueous phase was transferred to new Eppendorf tubes and supplemented with 500 µL of isopropyl alcohol (BioShop, ISO920.500). The samples were thoroughly mixed, incubated for 15 min at RT and centrifuged for 15 min at 4 °C by 12 000 rpm. The aqueous phase was discarded, and the pellet was washed with 1 mL of 75% ethanol (POCH Basic, Gliwice, Poland, BA6480111) and centrifuged for 5 min at 4 °C by 7600 rpm. The ethanol was discarded and the dried pellet dissolved in Rnase/Dnase water (BioShop, WAT333.500). The concentration and purity of the obtained RNA was measured on a Synergy LX microplate reader (BioTek) using the Take3 Micro-Volume Microplate (BioTek, Wnooski, VT, USA).

The level of 8-hydroxyguanosine (8-OHG) was quantified using the OxiSelect^TM^ Oxidative RNA Damage ELISA-8OHG Quantification Kit (Cell Biolabs, San Diego, CA, USA, STA-325). The preparation of samples included digestion of 10 µg of RNA with 20 U of Nuclease S1 (Bio Shop, Burlington, Canada, NUC333.50) for 2 h and with 10 U of alkaline phosphatase from bovine intestinal mucosa (Sigma-Aldrich, St Louis, MO, USA, P6774-2KU) for a further 1 h carried out at 37 °C. The subsequent steps were conducted according to the manufacturer’s manual. The absorbance at λ = 450 nm was measured on a Synergy LX Microplate Reader (BioTek, Winooski, VT, USA). The analyses were performed in three biological repetitions.

### 2.8. Quantification of Cd

For the quantification of Cd content, the roots of the soybean seedlings were thoroughly washed with distilled water, cut off and dried for three days in 60 °C. Then, the samples were sent for analysis using ICP-OES to a commercial company (Scallad, Poznań, Poland).

### 2.9. Statistical Analysis

The data were analyzed through -two-way analyses of variance (ANOVA) using SAS software (version 9.2), by *p* < 0.05.

## 3. Results

To confirm Cd uptake by the seedlings and assess possible interaction of MitoTEMPO with metal absorption, the level of Cd was quantified using inductively coupled plasma—optical emission spectrometry (ICP-OES). Unsurprisingly, the content of Cd in the roots of the control seedlings was nearly undetectable (Figure 1). In the case of the seedlings treated with CdCl_2_, metal accumulation was in correlation with the applied concentration—in the roots of seedlings treated with a lower Cd concentration (10 mg/L), the level of this metal reached nearly 200 µg/g DW; while in the roots of seedlings treated with a higher concentration (25 mg/L), it exceeded 700 µg/g DW. In the case of MitoTEMPO treatment, a significant increase in Cd uptake was observed in the treatment with lower metal concentration, while a decrease in Cd content was observed in the treatment with higher metal concentration

Metal treatment resulted in a dose-dependent inhibition of seedling growth. A significant reduction in the length of the seedlings roots and decreased roots fresh weight was observed in both applied metal concentrations. On the other hand, treatment with the mitochondria-specific antioxidant, MitoTEMPO, had no significant effect on the described growth parameters (Figure 2A,B). Roots shortening was the most visible Cd-dependent change in the seedlings phenotype (Figure 2C). In addition, exposure to this metal caused browning of the roots. Treatment with MitoTEMPO had no effect on the morphology of the seedlings.

Cadmium toxicity is frequently related to the development of oxidative stress. In the present study, the 48 h long exposure to Cd resulted in the increase in the level of common oxidative stress markers—superoxide anion (Figure 3A), hydrogen peroxide (Figure 3B) and lipid peroxidation (Figure 3C) in seedling roots. On the other hand, exposure to the metal did not affect the level of protein carbonylation in the roots (Figure 3D). To assess the possible involvement of mitochondria-derived ROS in the development of Cd-dependent oxidative stress, the mitochondria targeting antioxidant, MitoTEMPO, has been applied. Treatment with MitoTEMPO diminished Cd-dependent accumulation of superoxide anion and lipid peroxidation products (TBARS) observed in response to the higher Cd concentration (25 mg/L) but did not affect the level of hydrogen peroxide nor the level of carbonyl groups in the proteins (Figure 3A–D).

An elevated 8-hydroxyguanosine (8-OHG) level in the RNA is a recently described marker of short-term metal stress in plants [23,24]. The results of the present study showed that treatment with Cd resulted in a significant increase in the level of 8-OHG in total RNA isolated from seedling roots (Figure 4). This phenomenon was observed only in response to the application of lower Cd concentration (10 mg/L). Simultaneous treatment with MitoTEMPO caused a decrease in the 8-OHG level. However, the observed change was significant only by *p* = 0.07.

## 4. Discussion

Accumulation of ROS is one of the most common plant responses to Cd treatment [2,3]. These molecules are indispensable for proper functioning of, among other processes, signaling and gene regulatory elements [6]. However, their excess is toxic and leads to oxidative damage of cellular components including membrane lipids, proteins and nucleic acids [5]. Although Cd-dependent ROS accumulation in plants is well evidenced, there are still many open questions. For instance, there is very limited information on the sources of Cd-dependent ROS generation. Studies point to NAPDH oxidase and mitochondria as crucial ROS-producing sites under Cd stress [8,10,11].

In the case of mitochondria, the alterations in electron transport chains might lead to oxygen reduction and formation of reactive species, wherein complexes I and III are considered the main ROS-generating sites. Unfavourable conditions frequently disturb mitochondria functioning leading to an increase in ROS production. In general, mitochondria are recognized as important stress sensors and signalling hubs, as their altered functioning leads to global changes in gene expression and cell metabolism [25]. Disturbances in mitochondria ultrastructure and functions have been reported also in plant response to Cd. In rice seedlings, exposure to this metal led to disintegration of mitochondria membrane and a decrease in the activity of electron transport complexes I, II and IV [26]. In the roots of soybean plants, Cd stimulated KCN-insensitive but inhibited KCN-sensitive respiration [27]. In turn, in pea embryonic axis, Cd caused a decrease in total protein level in mitochondria with a simultaneous increase in thiol content, indicating alterations in oxidative status [28].

The aim of the present study was evaluation of the engagement of mitochondria-derived ROS in soybean response to short-term Cd stress. It should be highlighted that the experiments were conducted on young seedlings, with no developed leaves, which were cultivated in the dark. Therefore, in the present experimental conditions, chloroplasts could be excluded as the ROS-generating site. To elucidate the possible role of mitochondria-derived ROS in seedlings reaction to Cd mitochondria targeting antioxidant, MitoTEMPO, has been applied. This compound was applied in numerous studies on animal models showing its ability to supress ROS generation in mitochondria and exhibit protective effects in the case of cancer and neurodegenerative and cardiovascular disorders [12,13,14,15,16,17,18]. However, so far its application in plant biology is limited to solely one study carried out on elm seeds [19].

The results of the present study showed that even relatively short-term (48 h) treatment with Cd hampers seedling growth (Figure 2) and leads to the generation of O_2_^·−^ and H_2_O_2_ in the seedling roots accompanied by augmented lipid peroxidation (Figure 3). These are common symptoms of Cd stress observed in numerous plant species including Arabidopsis, rice, wheat, maize, barley, alfalfa, pea, lupine, spinach, tomato and sugar beet, reviewed in [2,3]. In accordance with the present study, ROS accumulation is in general indicated as an early reaction to Cd; in some cases, observed within the first hour of metal exposure [8,11,29,30].

Notably, in the present study, treatment with a lower applied Cd concentration (10 mg/L) also resulted in elevated levels of the most common oxidative modification of RNA, 8-hydroxyguanosine (8-OHG) (Figure 4). The information on transcripts oxidation in plants is still very limited, with less than 10 articles published overall (Scopus database search with terms “RNA oxidation” and “plant*”, date of access 19 October 2022). However, rapid induction of 8-OHG formation in RNA has been observed in plants response to metals [23,24]. In the cited studies, 8-OHG accumulation occurred in earlier periods (within the first 3 h of treatment) and reached the control levels after 24 h. Thus, the obtained results showing 8-OHG induction also after 48 h of treatment indicate fluctuations in Cd-dependent 8-OHG accumulation, possibly corresponding to distinct waves of ROS production [8]. The role of 8-OHG formation in response to metals is still undiscovered. The early 8-OHG formation observed only in response to lower Cd concentrations may suggest that this reaction is not necessarily a symptom of oxidative stress but might be engaged in regulatory reactions. In fact, it has been evidenced in sunflower and wheat seeds that 8-OHG formation constitutes a selective process leading to decreased biosynthesis of specific proteins required for alleviation of seed dormancy [31,32]. Additionally, in Arabidopsis high 8-OHG levels induced by ^1^O_2_ accumulation were accompanied by the suppression of the translation process [33].

To assess the putative involvement of mitochondria-derived ROS in the observed Cd-dependent oxidative events the mitochondria targeting antioxidant, MitoTEMPO, was applied. Surprisingly, the results show that MitoTEMPO affected Cd absorption in a concentration-dependent manner (Figure 1). In the case of lower metal concentration (10 mg/L), an increase in Cd content was noted in the roots of seedlings simultaneously exposed to MitoTEMPO. Notably, an opposite effect was observed in the seedlings treated with higher Cd concentration (25 mg/L). In this case, simultaneous application of MitoTEMPO resulted in hampered Cd uptake. Cd is frequently called an “opportunistic hitchhiker” as it passes into the cells through transporters predestined for essential minerals such as Ca^2+^, Zn^2+^ and Fe^2+^. The main transporters engaged in Cd uptake, translocation and accumulation include heavy metal-transporting ATPases (HMAs), resistance-associated macrophage proteins (Nramps), zinc regulated/iron regulated transporter (ZRT/IRT)-related proteins (ZIP) and yellow stripe-like proteins (YLS) [34]. It has been recently suggested that the activity of transporters and thus mineral uptake can be regulated through retrograde signalling, wherein ROS are indicated as the key retrograde signalling elements [35,36]. Therefore, we hypothesize that the observed MitoTEMPO-dependent changes in Cd content may result from modulation of retrograde signalling leading to changes in expression of genes encoding proteins involved in metal transport (e.g., transporters, chelators) on the transcriptional or posttranscriptional level. It is possible that in lower Cd concentrations signaling mediated by mitochondria-derived ROS leads to activation of defense genes and hampered metal uptake. On the other hand, in higher Cd concentrations, ROS produced in mitochondria mediate increased lipid peroxidation, which results in general disfunction of membranes and, in consequence, increased Cd absorption. Further detailed analysis would be needed to confirm the hypothesis. It would be also interesting to examine if MitoTEMPO modulates uptake of other non-essential as well as essential elements.

The results of the present study also showed that treatment with MitoTEMPO modulated the oxidative response of soybean seedlings to Cd. Simultaneous application of this antioxidant diminished Cd-dependent accumulation of O_2_^·−^ and lipid peroxidation products (Figure 3). Studies on tobacco suspension cells pointed to mitochondria as the main ROS source in short-term Cd exposure. The impairment of the mitochondrial electron transport chain through application of rotenone with thenoyltrifluoroacetone, myxothiazole or antimycin A resulted in a 70–90% decrease in the level of superoxide anion [8]. Similarly, microscopic observation of Arabidopsis protoplasts subjected to Cd revealed that ROS accumulation co-localized with mitochondria. This was accompanied by mitochondria aggregation and impaired movement [11]. The results of the present study also indicate that mitochondria might be engaged in Cd-dependent ROS over-production. However, it should be highlighted that it is not clear if the observed effect is dependent on the antioxidant action of MitoTEMPO or decreased Cd levels.

On the other hand, the lower level of 8-OHG in the roots of seedlings simultaneously exposed to MitoTEMPO and Cd when compared with the roots of seedlings treated solely with Cd, suggest that mitochondria-derived ROS are involved in RNA oxidation. In this case, the observed effect is not related to decreased Cd uptake, as in lower concentration (10 mg/L), MitoTEMPO induced absorption of the metal.

## 5. Conclusions

Exposure of seedlings to Cd resulted in development of common toxicity symptoms including impeded growth, accumulation of reactive oxygen species and intensified lipid peroxidation. In addition, Cd at lower concentrations (10 mg/L) induced RNA oxidation, reflected by augmented levels of 8-hydroxyguanosine (8-OHG). Application of the mitochondria-targeting antioxidant, MitoTEMPO, resulted in modified Cd uptake by soybean seedlings suggesting that mitochondria-derived ROS are engaged in regulation of metal uptake. Notably, the observed effect varied between applied Cd concentrations—an increase in metals uptake was noted by treatment with lower Cd concentration, while a decrease was observed by treatment with a higher concentration. The observed changes in Cd uptake were accompanied by diminished accumulation of superoxide anion and lipid peroxidation products.

## Figures and Tables

**Figure 1 antioxidants-11-02099-f001:**
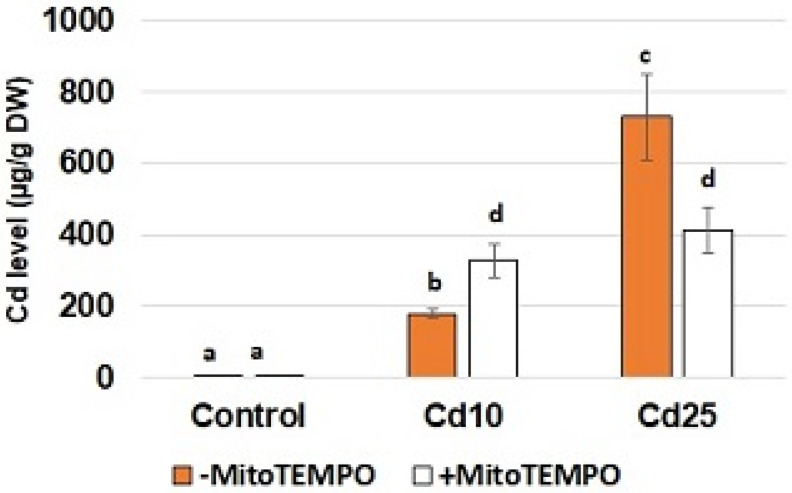
The level of Cd in control soybean seedlings and seedlings exposed to Cd at the concentration 10 and 25 mg/L. White bars represent soybean seedlings additionally treated with 20 nM MitoTEMPO in relation to the untreated seedlings represented by orange bars. The results are mean of 5 experiments ± SE. Statistically significantly different results are marked with different letters.

**Figure 2 antioxidants-11-02099-f002:**
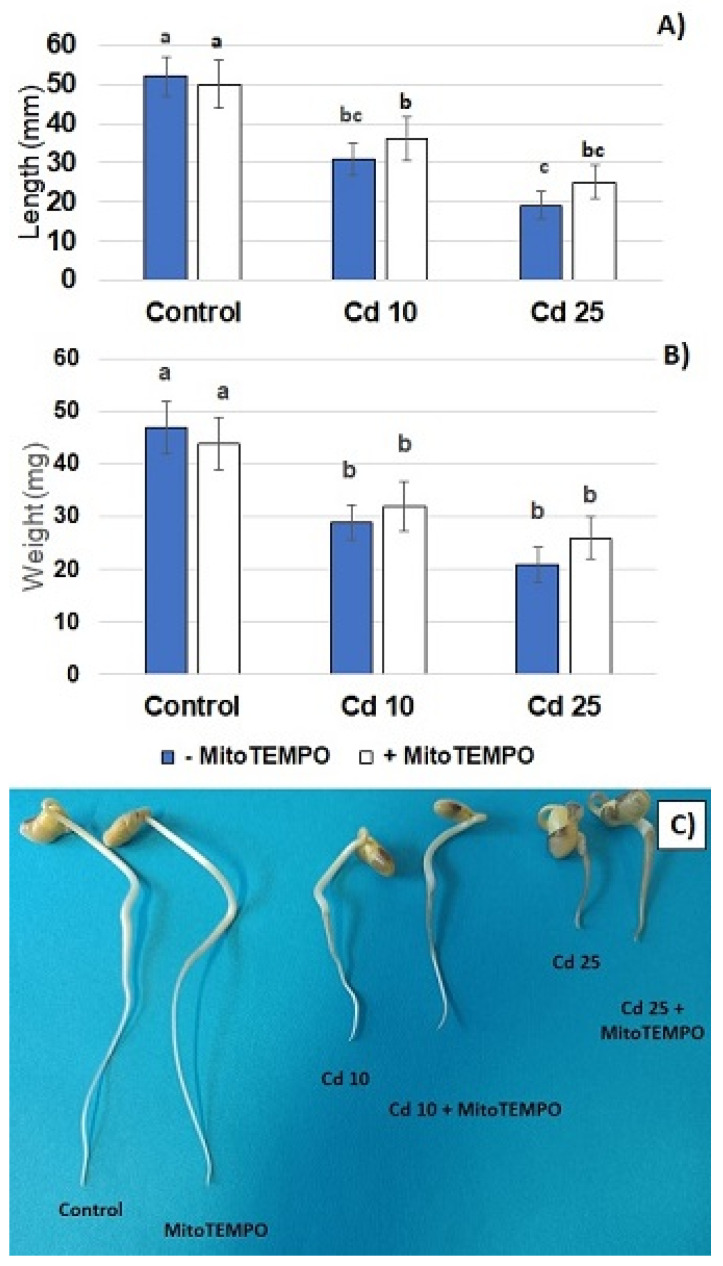
Roots length (**A**) and root fresh weight (**B**) of control soybean seedlings and soybean seedlings exposed for 48 h to CdCl_2_ with Cd at the concentration 10 and 25 mg/L. White bars represent soybean seedlings additionally treated with 20 nM MitoTEMPO in relation to the untreated seedlings represented by blue bars. The results are mean of 5 experiments ± SE. Statistically significantly different results are marked with different letters. The morphology (**C**) of control soybean seedlings and soybean seedlings treated for 48 with CdCl_2_ with Cd at the concentration 10 or 25 mg/L (Cd 10 and Cd 25, respectively) and/or with mitochondria-specific antioxidant, MitoTEMPO (Cd10 + MitoTEMPO and Cd25 + MitoTEMPO).

**Figure 3 antioxidants-11-02099-f003:**
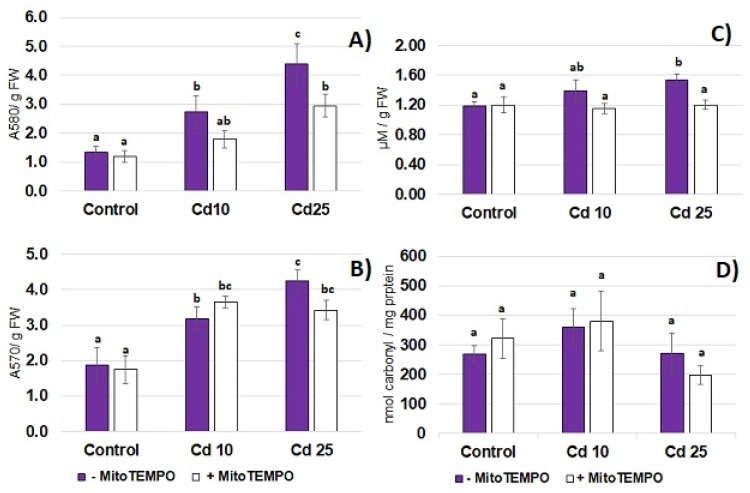
The level of superoxide anion (**A**) and hydrogen peroxide (**B**) thiobarbituric reactive substances (TBARS) reflecting lipid peroxidation (**C**) and carbonyl groups reflecting protein carbonylation (**D**) in the roots of control soybean seedlings and soybean seedlings exposed for 48 h to CdCl_2_ with Cd at the concentration 10 and 25 mg/L. White bars represent soybean seedlings additionally treated with 20 nM MitoTEMPO in relation to the untreated seedlings represented by purple bars. The results are mean of 4–6 experiments ± SE. Statistically significantly different results are marked with different letters.

**Figure 4 antioxidants-11-02099-f004:**
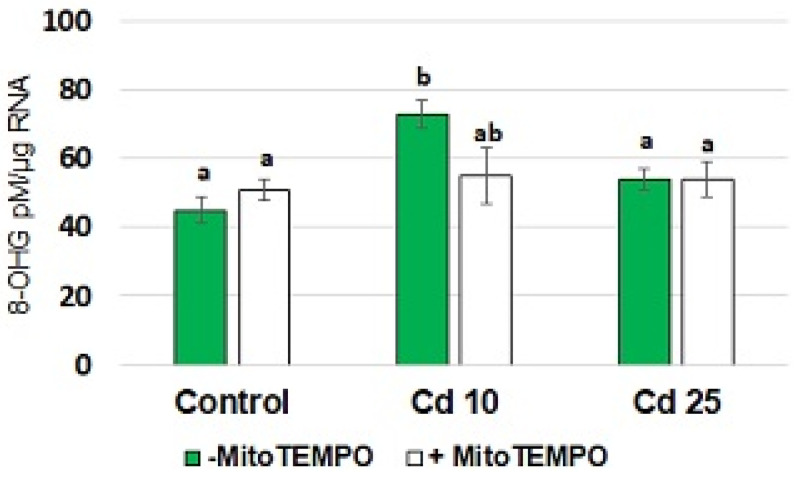
The level of 8-hydroxyguanosine (8-OHG) in total RNA isolated form the roots of control soybean seedlings and soybean seedlings exposed for 48 h to CdCl_2_ with Cd at the concentration 10 and 25 mg/L. White bars represent soybean seedlings additionally treated with 20 nM MitoTEMPO in relation to the untreated seedlings represented by green bars. The results are mean of 3–5 experiments ± SE. Statistically significantly different results are marked with different letters.

## Data Availability

The data presented in this study are available on request from the corresponding author.

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
