# Peer review of "Mitochondria Specific Antioxidant, MitoTEMPO, Modulates Cd Uptake and Oxidative Response of Soybean Seedlings"

_antioxidants, 2022, doi:10.3390/antiox11112099_

Round 1
Reviewer 1 Report
Chmielowska-Bąk and colleagues showed well written paper with fine data and good interpretation. However, a few minor queries need to be addressed before publication.
In my opinion the Introduction section is too generally, I think background of the performed investigations should be improved.
154 l. 8-OHG – should be explained
181-182 l. should be alfa =0.05 or p<0.05
Figures – number of repetition should be given
231 l. – correct pleas “p = 0.07”
Author Response
Dear Reviewer,
We would like to thank for valuable comments, which contributed to the improvement of the manuscript. Below we address all the comments in detail. We have marked all added/modified text in blue colour and all deletions in red colour.
Comments and Suggestions for Authors
Chmielowska-Bąk and colleagues showed well written paper with fine data and good interpretation. However, a few minor queries need to be addressed before publication.
In my opinion the Introduction section is too generally, I think background of the performed investigations should be improved.
Each paragraph of the introduction has been significantly expanded - all added fragments are marked in blue colour.
154 l. 8-OHG – should be explained
The explanation has been added (marked in blue colour).
181-182 l. should be alfa =0.05 or p<0.05
The equation has been changes (marked in blue colour)
Figures – number of repetition should be given
The number of repetitions (experiments) has been added in the description of all of the figures (marked in blue colour).
231 l. – correct please “p = 0.07”
The equation has been corrected (marked in blue colour).
Reviewer 2 Report
The manuscript “Mitochondria specific antioxidant, MitoTEMPO, modulates Cd uptake and oxidative response of soybean seedlings” investigated the roles of a specific antioxidant derived from mitochondria, MitoTEMPO, on Cd uptake and oxidative responses of soybean seedlings. The manuscript seems interesting, although some shortcomings exist in the current form. I would like to suggest some concerns for the present version.
Major
1. In my opinion, the statistical analysis method can be changed or added. Multiple range comparison test could not fully reflect the results shown in the figures. I think student’s test would give more information between groups with or without MitoTEMPO.
2. The lowercase letters in the figures are not uniform. In some cases, the largest number was marked “a”, whereas some times the lowest was given “a” too.
3. The tense of the manuscript should be uniform.
4. The reference part is too short. To my knowledge, there are some newly published papers aimed at Cd accumulation regulation.
Minor
1. Line 11: studies?
2. Lines 14:showed
3. Lines 41 & 45: dependent? Please confirm
4. Line 89: here, the authors just described 5 treatments, where is the remaining one?
5. Lines 93-94: How much time did the treatment perform? 48 hours? Why transferred?
6. Line 177: send?
7. Lines 190: Is there any other information here?
Author Response
Dear Reviewer,
We would like to thank for valuable comments, which contributed to the improvement of the manuscript. Below we address all the comments in detail. We have marked all added/modified text in blue colour and all deletions in red colour.
The manuscript “Mitochondria specific antioxidant, MitoTEMPO, modulates Cd uptake and oxidative response of soybean seedlings” investigated the roles of a specific antioxidant derived from mitochondria, MitoTEMPO, on Cd uptake and oxidative responses of soybean seedlings. The manuscript seems interesting, although some shortcomings exist in the current form. I would like to suggest some concerns for the present version.
Major
- In my opinion, the statistical analysis method can be changed or added. Multiple range comparison test could not fully reflect the results shown in the figures. I think student’s test would give more information between groups with or without MitoTEMPO.
We have rechecked the results using ANOVA test and changes the description in the material and methods section (marked in blue).
- The lowercase letters in the figures are not uniform. In some cases, the largest number was marked “a”, whereas some times the lowest was given “a” too.
We have marked control with an “a” in all of the graphs to facilitate the comparison between the control and other variants. However, if it is necessary we can of course change the designation of the bars on the graphs.
- The tense of the manuscript should be uniform.
The tense of the manuscript has been unified throughout the manuscript (marked in blue colour).
- The reference part is too short. To my knowledge, there are some newly published papers aimed at Cd accumulation regulation.
The references list has been significantly expanded, including articles on Cd uptake. All the added references are marked in blue colour.
Minor
- Line 11: studies?
The term has been changed to “studied” (marked in blue colour).
- Lines 14:showed
The word has been corrected (marked in blue colour).
- Lines 41 & 45: dependent? Please confirm
Indeed, the word should be “depended”. We have changes the term in both line (marked in blue colour).
- Line 89: here, the authors just described 5 treatments, where is the remaining one?
Thank you for the comment. Indeed, we have missed one of the variants. The variant description has been added in the paragraph (marked in blue).
- Lines 93-94: How much time did the treatment perform? 48 hours? Why transferred?
We have highlighted that the treatment was carried out for 48 hours.
- Line 177: send?
Indeed, should be sent. The word has been changes (marked in blue colour).
- Lines 190: Is there any other information here?
We did not add any other information in the line.
Reviewer 3 Report
The manuscript entitled “Mitochondria specific antioxidant, MitoTEMPO, modulates Cd uptake and oxidative response of soybean seedlings” aims to address the effect of mitochondria specific antioxidant, MitoTEMPO, on the response of soybean seedlings to short term Cd stress.
I think this research has certain innovations. Further, I think that the topic content of this manuscript is also suitable for the international audiences of antioxidants.
I encourage the authors to review and modify carefully the manuscript and after checking the below points I am pretty sure it should be accepted for publication:
- The materials and methods section needs further precisions (particularly “the Cultivation and treatment procedures” section).
- L86: “48 hours” of germination.
- Results section: Overly descriptive.
- This is a short term Cd stress application, a short term mitochondria targeting antioxidant, MitoTEMPO application and the experiments were carried out on very young seedlings, with no developed leaves, which were cultivated in the dark. The authors in the discussion section must take into consideration these experimental conditions.
- Discussion section: Not enough depth for the discussion. Please improve the readability as far as possible. Further, the authors must fully elaborate the most significant results. The depiction in the discussion section must be closely connected to the results of this manuscript and the results of previous investigators.
Author Response
Dear Reviewer,
We would like to thank for valuable comments, which contributed to the improvement of the manuscript. Below we address all the comments in detail. We have marked all added/modified text in blue colour and all deletions in red colour.
The manuscript entitled “Mitochondria specific antioxidant, MitoTEMPO, modulates Cd uptake and oxidative response of soybean seedlings” aims to address the effect of mitochondria specific antioxidant, MitoTEMPO, on the response of soybean seedlings to short term Cd stress.
I think this research has certain innovations. Further, I think that the topic content of this manuscript is also suitable for the international audiences of antioxidants.
I encourage the authors to review and modify carefully the manuscript and after checking the below points I am pretty sure it should be accepted for publication:
- The materials and methods section needs further precisions (particularly “the Cultivation and treatment procedures” section).
The material and methods section has been carefully read and the applied methods more detailly described (marked in blue colour).
- L86: “48 hours” of germination.
The term has been changes in the manuscript (marked in blue colour).
- Results section: Overly descriptive.
The results section has been rewritten and expanded to make it less descriptive. All modifications are marked in blue colour.
- This is a short term Cd stress application, a short term mitochondria targeting antioxidant, MitoTEMPO application and the experiments were carried out on very young seedlings, with no developed leaves, which were cultivated in the dark. The authors in the discussion section must take into consideration these experimental conditions.
- Discussion section: Not enough depth for the discussion. Please improve the readability as far as possible. Further, the authors must fully elaborate the most significant results. The depiction in the discussion section must be closely connected to the results of this manuscript and the results of previous investigators.
We have majorly revised the discussion section, which hopefully improved its readability, highlighted the applied conditions relating them to studies of other investigators and highlighted the most important obtained results. All added fragments are marked in blue colour, while all deletions are marked in red.